# Predicting Routine Object Usage for Proactive Robot Assistance

**Maithili Patel**
Georgia Institute of Technology
maithili@gatech.edu

**Aswin Prakash**
Georgia Institute of Technology
aprakash88@gatech.edu

**Sonia Chernova**
Georgia Institute of Technology
chernova@gatech.edu

**Abstract:** Proactivity in robot assistance refers to the robot's ability to anticipate user needs and perform assistive actions without explicit requests. This requires understanding user routines, predicting consistent activities, and actively seeking information to predict inconsistent behaviors. We propose SLaTe-PRO (Sequential Latent Temporal model for Predicting Routine Object usage), which improves upon prior state-of-the-art by combining object and user action information, and conditioning object usage predictions on past history. Additionally, we find some human behavior to be inherently stochastic and lacking in contextual cues that the robot can use for proactive assistance. To address such cases, we introduce an interactive query mechanism that can be used to ask queries about the user's intended activities and object use to improve prediction. We evaluate our approach on longitudinal data from three households, spanning 24 activity classes. SLaTe-PRO performance raises the F1 score metric to 0.57 without queries, and 0.60 with user queries, over a score of 0.43 from prior work. We additionally present a case study with a fully autonomous household robot.

**Keywords:** Proactive Assistance, User Routine Understanding, Human Activity Anticipation, Robot Learning

## 1 Introduction

Proactive assistive robots provide support for human user activities by monitoring user actions, identifying opportunities for supporting the user's objective, and performing supportive actions without explicitly being asked. Incorporating elements of proactive assistance has been proposed as a key principle for effective human-robot interaction [1], and studies have shown that users prefer proactive assistance over always having to ask for help in longitudinal interactions with robots [2, 3]. Prior work has considered assistance at two different time scales: short-term assistance based on the user's current action (e.g., handing the next tool for an assembly task) [4, 5], and longitudinal assistance, in which the robot must anticipate the user's needs over long time horizons (e.g., setting out breakfast before the user comes into the kitchen) [6].

In this work, we consider the problem of *longitudinal proactive assistance*, in which the robot learns patterns in user behavior from observations of a wide range of household tasks, and then provides assistance by fetching objects prior to being asked. Longitudinal assistance is a challenging problem due to the inherent stochasticity of human behavior – at any given time of day, a person may engage in a wide variety of activities or interact with many objects. The leading dataset for modeling proactive assistance, HOMER [6], crowdsourced different patterns of user routines and obtained models in which users were engaged in one of 3 activities on average, and up to 9 activities at certain times of the day.

Computationally, proactive assistance can be modeled by considering object-object relation frequencies [7], periodic routines [8], or through spatio-temporal object tracking [6]. Our work is particularly inspired by Spatio-Temporal Object Tracking (STOT) [6], which outperforms other prior methods using a generative graph neural network to learn a unified spatio-temporal predictive model of object dynamics from temporal sequences of object arrangements. The resulting model performed well on more consistent user routines, such as *using a plate for dinner*, but was unable to predict less consistent activities, such as *socializing*. A key limitation of STOT is that it utilized only object information for proactivity cues (e.g., which objects the user picked up and moved) and did not consider the underlying high level activity label for the user's actions.

7th Conference on Robot Learning (CoRL 2023), Atlanta, USA.

In this work, we contribute Sequential Latent Temporal model for Predicting Routine Object usage (SLaTe-PRO), which models temporal evolution of user activities by incorporating observations obtained in object and activity domains. SLaTe-PRO improves upon STOT by i) combining object and user action information, and ii) conditioning object usage predictions on the past history of user observations in addition to the current observation, leading to a significant improvement in proactivity performance. We further characterize human activities by their difficulty with respect to proactive assistance, and show that temporal consistency of activities plays a key role in enabling effective proactive assistance. Importantly, our analysis shows that some activities are so inconsistent and lacking in contextual cues that the robot can do no better than a random guess based on their likelihood. To address such cases, we introduce a user interaction component that enables the robot to make a limited number of daily user queries to inform assistance decisions.

We evaluate SLaTe-PRO on three households from the HOMER dataset, which incorporates models of 24 routine activities, and compare performance against prior state-of-the-art, STOT. Our results show that, under the same operating conditions as assumed in [6], in which no activity recognition data is available about the user, SLaTe-PRO outperforms STOT, raising F1 score from $0.43$ to $0.52$. With the added option to perform activity recognition to detect the user's current activity, SLaTe-PRO achieves a further improvement in F1 score to $0.57$. Finally, enabling the robot to pair SLaTe-PRO with a limited number of user queries leads to an F1 score of $0.60$, for the most significant improvement over STOT. We present detailed performance results in simulation, and then present a case study and description of a fully autonomous household robot.

## 2   Prior Work

Key elements in addressing the problem of proactive assistance include recognizing activities being executed by the user, modeling temporal patterns in observed activity representations, and finally interacting with the user to obtain clarifications.

**Activity recognition** has been explored through the use of camera data [9, 10, 11, 12], smart-home sensors [13, 14, 15, 16, 17], and wearable devices [18, 19]. In many settings, tracking the interaction of the user with various objects has been used to improve activity recognition [20, 21, 22], goal inference [4] and temporal routine tracking [6]. Recent work on smart-home systems for activity recognition [13] have achieved ~80% accuracy on recognizing activity labels at a coarse-grained level, such as breakfast, washing dishes, work in office, etc. Activity labels can provide useful cues about object needs, or lack thereof. For example, greeting guests might indicate the need to serve refreshments, while leaving the house suggests no need for anything. We seek to utilize such activity recognition labels in combination with object-centric observations to create a representation of user activities, which can support predictive modeling of their routines in object space.

Long-horizon proactivity requires a **temporal model** of the user's behavior in addition to understanding the activities independently. To represent temporal patterns in user routines we use latent space sequence models, which have increasingly been used to model world dynamics [23, 24, 25]. These works utilize latent representations to capture dynamics of objects resulting from physics in the environment, whereas we seek to capture the effect of user's activities and their effect on how objects move between different locations (e.g. cabinet, sink, etc.) around a large space, such as a whole household. Models of user behavior at longer timescales have been studied in the context of predicting occupancy and traffic [26, 27], and more recently towards anticipating object usage for proactive assistance [6], but they remain specific to their respective domain, by modeling a sequence over data represented in that domain. In contrast, our model can combine information obtained in various forms into a unified latent space and use that space to make predictions.

Finally, **user interaction** is necessary in proactive assistance, to address inevitable predictions errors resulting from aleatoric uncertainty in the user's daily life. Our proposed solution for seeking user feedback derives from ideas of information gain, which have been used in prior work to plan active actions towards searching and mapping unexplored regions [28, 29, 30, 31], to plan actions towards improving world models for reinforcement learning [32], to actively query labels to improve classification performance [33], and to find objects in clutter [34]. Verbal clarifications have been explored towards refining goal specifications [35, 36, 37] or navigation instructions [38]. While these works seek to utilize clarifications in natural language to refine user commands in the same space, we seek to clarify our model's predictions of the user's activities. Natural language expressions based on robot's internal inferences have been explored towards explaining the robot's actions to promote

explainability and transparency [39, 40]. In contrast to promoting the user's understanding of the robot's inferences, we seek to obtain information that improves the robot's predictions.

## 3 Problem Formulation

Our work builds on the formulation of *proactive assistance through object relocations* introduced in [6]; we extend the problem formulation to incorporate activity recognition data and to incorporate user interaction. An environment consists of a set of objects $\mathcal{O} = \{o_i\}$, and locations $\mathcal{L} = \{l_i\}$, and a human agent that takes actions which lead to the movement of objects from one location to another. Note that objects can also serve as locations for other objects (e.g. spoon being on a plate), in which case the object exists in both $\mathcal{O}$ and $\mathcal{L}$. The state of the environment $G_t$, at time $t$, consists of a set of object-location pairs $\{(o_i, l_i)\}$, and can be fully observed by the robot.

At any given time, the robot's classification of human agent's activity is represented as $a_t \in \mathcal{A}$, one of a predefined set of activities, where *unknown* $\in \mathcal{A}$ represents all activities not known to, or not recognized by, the robot. Over any given time period $\Delta t$, the human agent performs actions as part of their activity, causing the environment to transition to $G_{t+\Delta t}$, and the difference between states $G_t$ and $G_{t+\Delta t}$ can be represented as the set of objects that move to a new location within that timestep $\Delta G_{t:t+\Delta t} = \{(o_i, l_j) | (o_i, l_j) \notin G_t, (o_i, l_j) \in G_{t'}, \text{s.t. } t < t' < (t + \Delta t)\}$. Additionally, the user can provide input to the robot by specifying their intended activity or object usage through $u_t$. In our work, we enable the robot to query the user about their upcoming activities (e.g., "will you be having fruit for breakfast?"), but unprompted user input can also be captured within $u_t$.

We formulate proactive assistance as consisting of two phases, an observation (training) phase and assistance phase. In the observation phase, the robot obtains sequential observations of the environment state $G_t$ and user actions $a_t$, which it uses to learn a predictive model of the user's behavior. In the assistance phase, given a history of environment states $G_{0:t}$, partially observed history of activity labels $a_{0:t}$, optional user input $u_t$, and time $t$, the robot predicts a set of object relocations $\mathcal{R} = \{(o_i, l_j)\}$ consisting of the objects $o_i$ that change locations in the predictive window of $t$ to $t + \Delta t$, along with their *first* new location $l_j$, so that it can move the object where needed.

## 4 Predicting Routine Object Usage

The aim of SLaTe-PRO[1] is to utilize environment observations $G_{0:t}$ and user activity recognition labels $a_{0:t}$, when available, to model user routines and predict future object movements $\Delta G_{t:t+\Delta t}$. We model all observations and predictions at discrete time steps, represented as unit length in our notation. We represent the environment as scene graphs with nodes representing objects $o_i$ and locations $l_i$ through one-hot vectors, and edges connecting objects to their respective locations. Activity labels $a_t$ are represented as one-hot vectors, and time-of-day as a vector of sines and cosines of predetermined frequencies $\tau(t)$ based on prior work [41]. Our proposed method encodes all available activity labels and object arrangement observations into a shared latent space, makes predictions in this space, and decodes them into object movements, as outlined in Figure 1. The latent space $X_t$ encodes more detailed activity information beyond what is captured by the activity label. For instance, the label *having breakfast* captures the high level activity, whereas the latent space might additionally capture which food is being eaten and what utensils are being used.

We learn **autoencoder models** for object arrangements and activity lables, based on transformers, graph neural networks, and feedforward MLPs. The object arrangement encoder captures environment changes from the scene graph pairs $f_{enc} : G_{t-1}, G_t, \tau(t) \rightarrow \hat{X}_t$, and is modeled as a transformer [42] with attention across all objects. The encoder takes as input object features, concatenated with the distribution of their previous and current locations, to capture object movements while preserving the context of unchanged object locations. The encoder applies self-attention across input encodings of all objects, followed by cross-attention conditioned on the time-of-day context, and finally max pooling across resulting object features to obtain a latent representation of overall change in the environment state. Conditioning on time-of-day helps the model contextualize object movements, e.g. using a cup for coffee or wine. The object arrangement decoder $f_{dec} : G_t, X_{t+1} \rightarrow p(G_{t+1})$, modelled using an edge-message-passing-based graph neural network proposed in prior work [6], generates a probabilistic scene graph representing object arrangements at a future time step from the current scene graph and conditioned on the latent vector. Note that instead of encoding and decoding the entire scene graph, our approach focuses on capturing changes in the environment through the latent vector. We encode graph pairs and condition our decoder on the

---

[1]The code is available publicly at https://github.com/Maithili/SLaTe-PRO

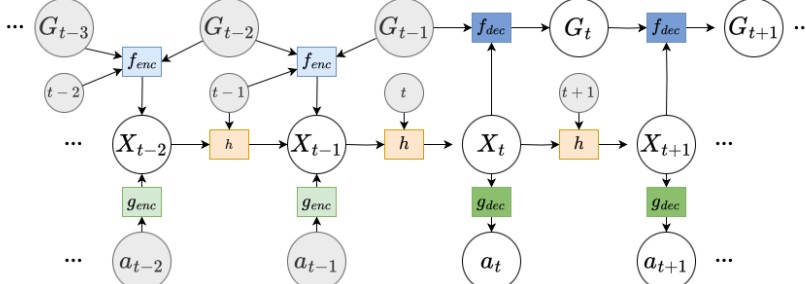

Figure 1: SLaTe-PRO consists of transformer-based encoder $f_{enc}$ and graph neural network based decoder $f_{dec}$ for object observations $G_t$, learned embeddings $g_{enc}$ and MLP-based decoder $g_{dec}$ for activity labels $a_t$, and a transformer-based predictive model $h$ over latent space $X_t$. These models together learn to predict the object arrangement and activity label at future time-steps. The variables in grey represent observed variables

previous graph, not requiring our model to remember locations of irrelevant objects. We represent the activity label encoder as a learnable set of embedding vectors $g_{enc} : a_t \rightarrow \hat{X}_t$ for each activity label, and the activity label decoder as a fully connected feedforward classifier $g_{dec} : X_t \rightarrow p(A_t)$.

Finally, we create a **latent dynamics model** $h : X_{0:t}, \tau(t) \rightarrow \hat{X}_{t+1}$ to learn the user's temporal routine in the shared latent space. This model predicts the next latent state given a history of latent states and current time-of-day using self-attention-based transformer encoder. We integrate absolute time vector $\tau(t)$ with the latent vectors by summation, similar to positional embeddings in the original transformer. Using the absolute time provides the model with the semantic context of time-of-day, in addition to the relative temporal sequence of the latent vectors.

We use several **training objectives** to train all components of our model simultaneously, using sequential observations of the environment and the user. We use reconstruction losses to train each autoencoder, specifically crossentropy losses for predicting the correct activity label, and the correct location of each object. We train latents obtained from either source to be similar through a contrastive loss, and combine latents obtained from both encoders by averaging. The latent predictive model is trained on this averaged latent through both, reconstruction losses on the decoded future graphs and actions, and contrastive loss between the predicted and encoded latent vectors. We use a latent overshooting loss to aid long-horizon predictions, as proposed in prior work [23], and find that for our model, observation overshooting provides very little benefit.

At **inference** time, the model must predict object relocations $\hat{\mathcal{R}}$ that will occur in a given $\delta$-step predictive horizon. For this we first encode observation histories $G_{0:t}$ and $a_{0:T}$, and average the encodings at every timestep to obtain a sequence of latent vectors $X_{0:T}$. We then employ the latent predictive model to predict latent vectors for a $\delta$-prediction horizon $\hat{X}_{t:t+\delta}$, and decode them into a sequence of object arrangement probabilities $p(G_{t:t+\delta})$ and activity labels $\hat{A}_{t:t+\delta}$. To predict relocations $\hat{\mathcal{R}}$ associated with each object's first movement, we use the location distribution of the time-step $t_{o_i}$ when the object $o_i$, currently at location $l_i$ is most likely to be at a different location $t_{o_i} = \text{argmin}_t \ p(o_i, l_i)$. By combining the location distributions of all objects, we can infer a probability distribution over object-location pairs $p_{reloc}(o_i, l_j)$, and infer relocations $\hat{\mathcal{R}} = \{(o_i, \hat{l}_j)\}$ as a set of objects $o_i$ that are predicted to move to locations $\hat{l}_j$. In a similar manner, we can infer a distribution over next activity prediction as the probability of each activity label at the time-step $t_a$ when an activity different from the current activity $a_0$ is most likely to start $t_a = \text{argmin}_t \ p_t(a_0)$.

## 5 Overcoming Stochastic User Behavior through Interactive Queries

The above framework is effective when future human behavior can be predicted from past observations. However, some human action choices are inherently more stochastic than others. In this section we discuss this occurrence and present our approach for generating interactive queries for proactive assistance.

### 5.1 Limitations to Computational Proactive Assistance

A model that predicts user needs is fundamentally limited by the stochastic nature of human behavior. Users may engage in some activities less consistently than others, such as choosing to eat out, cook dinner at home, or host a party on various nights. Alternately, users may perform the same activity using different objects, such as choosing between cereal, oatmeal or fruit for breakfast. In our analysis of the HOMER dataset, we found that for many such cases there are no observable be-

havioral cues that the robot can use to predict the user's actions. In such cases, the predictive model can do no better than chance, even when entirety of ground truth observations is made available. Unsurprisingly, as we report in the results section, performance of SLaTe-PRO drops by 26% on such less consistent activities compared to the overall dataset. In the section below, we describe an interactive approach for eliciting additional information from the user in response to robot queries relating to activities (e.g., "will you be having dinner soon?"), or object usage (e.g. "will you have cereal for breakfast today?").

## 5.2 Interactive Queries for Proactive Assistance

We rely on the learned predictive model to decide when an inconsistent behavior is likely to occur, and which query $q$ would elicit information $\mathcal{Q} : q \to u_t$ that best alleviates the uncertainty. Specifically, we use the predicted relocation distribution $p_{reloc}$ to focus on predictions that the robot is most uncertain about and which might provide useful assistive opportunities. We use information gain as a metric to decide when a query will be informative, and measure it through the expectation over the potential query responses $u_t$ of reduction in entropy of the relocation distribution $\mathcal{H}(p_{reloc})$. We use the predicted activity and object relocation distributions as the probability of query responses indicating the respective events, and calculate the expected information gain as

$\sum_{u_t} p(u_t)\big(\mathcal{H}(p_{reloc}(o_i, l_j)) - \mathcal{H}(p_{reloc}(o_i, l_j | u_t))\big).$

The robot can elicit query responses in two forms $u_t \in \{u_t^a, u_t^{o_i}\}$, by asking regarding an activity $a$, resulting in the activity that the user will do next, $u_t^a \in \mathcal{A}$, or regarding a particular object $o_i$, resulting in a binary response on whether it will be used in the predictive horizon, $u_t^{o_i} \in \{True, False\}$. We interpret the response to an activity query as the correct activity label at the time-step $t_a$ when a new activity is likely to start (similar to the next activity inference in Section 4). We encode the activity to obtain a vector in latent space $\tilde{X}_{t_a} = g_{enc}(u_t)$, which we combine with the predicted latent at that time-step through a weighted average. The object relocation distribution conditioned on the query response $p_{reloc}(o_i, l_j | u_t)$, is obtained by continuing the rollout with the corrected latent vector. For object queries, we interpret the user response as the object leaving from or staying in its current location at the time-step $t_{o_i}$ when it was most likely to move (similar to the relocation inference in Section 4). We obtain the conditioned object relocation distribution by correcting the latent vector and continuing the remaining rollout, similar to activity-based queries.

We compute the expected information gain from all potential query candidates, which includes the activity-based query, and all objects that do not have highly confident predictions. We exclude objects which the model is over 90% confident about to avoid unnecessary computational overhead. If the best query has an expected information gain of above a predefined threshold, then the robot will ask that query, and correct its predictions based on the response.

## 6 Evaluation

We create HOMER+[2], by modifying the HOMER dataset [6], which represents routines of individual households over several weeks. The original HOMER dataset contains 5 households with 22 activities, and focuses on capturing variations across households. Each activity is executed in a household using a single crowdsourced script (scripts may differ between households). We modify this dataset to instead focus on capturing realistic variations within each household. We adopt a script per activity from the original dataset, and add 1-3 variations for 17 out of 24 activities[3] to better emulate how humans perform the same activity in various ways, eg. sometimes having *cereal*, and other times having *oatmeal* for breakfast. The resulting behavior distribution more accurately reflects real world human behavior. Note that this variation challenges our model just as much as the baseline, if not more, as the robot observes the same activity label regardless of the variation being executed, but we believe this more accurately reflects the stochasticity of user routines. HOMER+ includes 3 households, with 24 activities, and 93 entities, including objects and locations.

We evaluate our model's predictions of object relocations $\hat{\mathcal{R}} = \{(o_i, \hat{l}_j)\}$ by comparing them against the expected set of relocations in the ground truth sequence $\mathcal{R} = \{(o_i, l_j)\}$ over metrics of recall $\frac{|\hat{\mathcal{R}} \cap \mathcal{R}|}{|\mathcal{R}|}$, precision $\frac{|\hat{\mathcal{R}} \cap \mathcal{R}|}{|\hat{\mathcal{R}}|}$, and F-1 score calculated based on them. We evaluate predictions over a $\delta$-step predictive window, independently for each time-step $t$, with a discretization of 10 minutes. We consider a predictive window of 30 mins, unless otherwise specified. All our evaluations are based

---

[2]HOMER+ is included with SLaTe-PRO code at `https://github.com/Maithili/SLaTe-PRO`

[3]We add *going to sleep*, *getting out of bed* and *taking a nap*, split *washing dishes* into activities associated with each meal, and combine *cleaning*, *kitchen_cleaning* and *vacuum_cleaning* into a single activity

on 10 days of evaluation data per household, which are unseen during training. All metrics are micro-averaged over the 10 days for each household and macro-averaged over the three households.

We compare our results against STOT, proposed in prior work [6]. STOT utilizes time as context to make one-step predictions on object arrangements, and iteratively does so for long-horizon predictions. The architecture and size of this model is the same as our object movement decoder. We train all models with 60 days of observations, independently for each household. The size of our latent vector and embeddings of both transformers are set to 16, and the hidden layer of our GNN decoder and STOT are set to 8 as in [6]. We train all models up to 1000 epochs with early stopping based on accuracy over predicting locations of moved objects, using Adam optimizer and a learning rate of $10^{-3}$. We use a threshold of 0.5 on information gain and 0.8 as the weight of latent correction when incorporating feedback.

## 7 Results
We present empirical results on the HOMER+ dataset, show the effect of inconsistency on performance, and how active queries help improve performance, especially over inconsistent behavior. Finally, we present a case study of SLaTe-PRO running on a physical robot system.

### 7.1 Predictive Performance against STOT
Figure 2a presents a comparison between variations of SLaTe-PRO against STOT. We see that in the absence of activity recognition (i.e., given exactly same inputs $G_t$), SLaTe-PRO (No Act) outperforms STOT by 20%. This improvement is due to the recurrent latent space prediction, which leverages the history of past states in addition to the current state. For example, if the user had breakfast a few hours ago, they are unlikely to have it again.

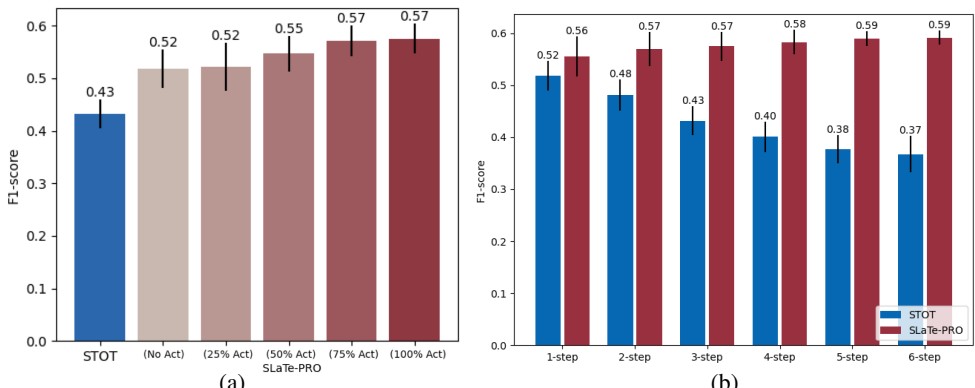

Figure 2: (a) SLaTe-PRO outperforms STOT with no activity labels, and steadily improves as more activities become available. (b) With 100% activity labels, SLaTe-PRO outperforms STOT across varying proactivity $\delta$-s

Next, we evaluate the benefit of activity labels on the performance of SLaTe-PRO. Modern activity recognition systems are not perfect, as discussed in Section 2, and achieve ~80% accuracy on datasets, and potentially lower in complex real world settings. Hence, we consider varying activity recognition performance, from none (No Act) to 100% availability of correct activity labels (Figure 2a). With 75% activity labels, SLaTe-PRO approaches peak performance, raising the F1 score from 0.43 to 0.57, showing little additional improvement with more activity labels. In Figure 2b, we analyze SLaTe-PRO with 100% activity labels across different predictive-$\delta$s, and demonstrate that it outperforms STOT, particularly for long-horizon predictions.

### 7.2 Effect of Behavioral Consistency
Next, we study how the model performance differs for consistent and inconsistent activities. We assess activity consistency using standard deviation $\sigma$ in start times, and categorize them into three groups: more consistent ($\sigma < 30$min), less consistent ($\sigma > 1$hr), and moderately consistent activities falling in between. For activities that occur multiple times a day, such as brushing teeth in the morning and night, we separately calculate the standard deviation per occurrence, by clustering using k-means with the average number of occurrences per day as the value for k. We then evaluate over objects that participate in activities in each of these categories separately. If an object is involved in multiple activities, we evaluate it based on the less consistent category. By splitting the dataset in this manner, across the three household datasets, 39% object movements fall in the more consistent category, 31% in the middle, and 30% in the less consistent category.

Predictably, the performance of SLaTe-PRO as well as STOT falls with decreasing consistency, as shown in Figure 3. The performance gap between the more consistent and less consistent activities in SLaTe-PRO with 100% activity availability is 0.22, and that for STOT is 0.12. We find the usage of toothpaste in one of our datasets as the most extreme case of routine usage, with a standard deviation of 5 mins. On that object alone, all methods achieve an F-1 score of 0.97.

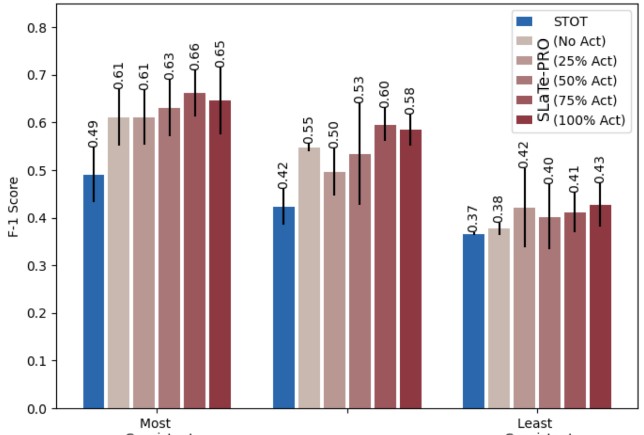

Figure 3: A steady drop in performance is observed across all methods from more consistent object usage to less.

## 7.3 Improvement from Queries

To address the impact of human behavior variability on predictive performance, we examine the effectiveness of active queries, particularly for inconsistent activities. For each prediction, the robot is allowed to ask a single query, and an oracle response is provided if a query is asked. We set a threshold of 0.5 information gain for every model to decide when a query should be asked. Figure 4 shows performance gains from the inclusion of active queries, with most significant improvements over the less consistent object usages, raising the F1-score from 0.43 to 0.49. The F1-score on the overall dataset improves from 0.57 to 0.6, while performance of STOT only improves by 0.02 overall as well as over inconsistent activities. Note that STOT can only ask object-based queries. SLaTe-PRO seeks feedback for about 18% of its predictions, while STOT asks 8% with the same threshold over information gain. Even if we encourage it to ask a similar number of queries as our model by reducing the threshold, we only see an improvement of 0.01. This indicates that STOT, despite adopting a conservative approach to avoid false object movements, fails to effectively represent uncertain predictions and seek useful feedback. We also find that our model does not ask queries pertaining to very sporadic activities, whereas STOT sometimes devotes unnecessarily many queries to such activities. For instance, in one of our datasets where the user tends to watch TV randomly over the day, STOT persistently asks the user about TV remote usage in the afternoon. In contrast, our model disregards this event and only intervenes in returning the remote. This allows our model to prioritize other, more relevant queries, while STOT misses opportunities. Our model focuses on other inconsistent activities which might lead to more informative results. For instance, if a user typically has dinner between 6pm and 8pm, our model asks about dinner plans to avoid prematurely preparing objects or delaying preparations and missing an assistive opportunity.

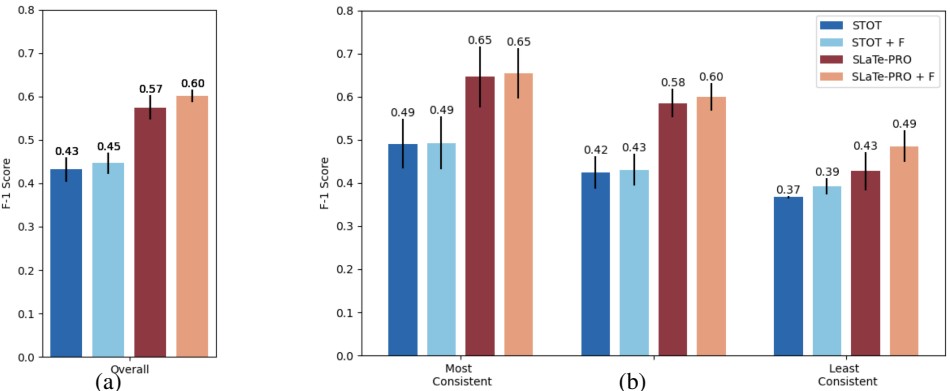

Figure 4: (b) Performance gains from queries are most significant for the less consistent object usages. (a) They also improve performance across the entire dataset. '+F' denotes using active queries with the method.

Additionally, our model is able to discern whether an object query or an activity query would be more useful at a given time. If our model is constrained to asking only object-based or only activity-based queries, we obtain an F1-score of 0.46, in either case, as opposed to 0.49 when the model can choose to ask any type of query. If the model is uncertain about the occurrence of the activity as a whole, it tends to ask activity-based queries, such as "Will you be socializing soon?", otherwise it seeks more specific information, such as "Will you be needing the coffee when you socialize?". For activities

wherein a particular object is consistently used, the model slightly favors object-based queries, such as asking about requiring oil for cooking dinner. Although both types of queries contain the same information, the object-based queries might be easier for the model to condition on, being in the same form as the desired output. While generally our model correctly picks the more informative query between activities and objects, it does sometimes go wrong, causing two kinds of failures. First, when the model overestimates its confidence in an activity label causing the negative response to an object use leading the model to predict another variation of the activity. For instance, if the model receives feedback of 'not using cereal' it may mistakenly predict the use of oatmeal because it is certain that the user will have breakfast, even when the user intends to engage in a different activity. On the other hand asking only about the activity label when underconfident leaves it to the robot's discretion to predict which variation the user prefers, still leaving room for errors.

## 7.4  Robot Validation

We demonstrate our proactive assistance system on a Stretch robot [43] in a household setting, portraying a morning routine, as shown in Figure 5 and in our video[4]. To infer a semantic scene graph representation of the environment from visual observations, we first reconstruct the scene with objects represented as meshes and bounding boxes, using Hydra [44][5], with dense semantic labels obtained from SparseInst [45]. We extract the object-location relations through heuristics based on bounding boxes, and use them to build and dynamically update a scene graph as objects move across the environment. The user's routine is demonstrated to the robot as first having breakfast, where they usually have an apple and coffee but sometimes eat cereal, and usually leave for work afterwards, but sometimes work from home. Scene graphs and activity labels obtained during the observation phase serve as training data for SLaTe-PRO. During the assistance phase, the robot acts on its object relocation predictions to assist proactively (Figure 5a). However, when the user chooses to follow their less common behavior, e.g. eating cereal (Figure 5b), the robot's assistive actions fail. The robot is able to correct such mistakes by actively querying the user about their intended object usage (apple or cereal), and activity (leaving or working from home) (Figure 5c).

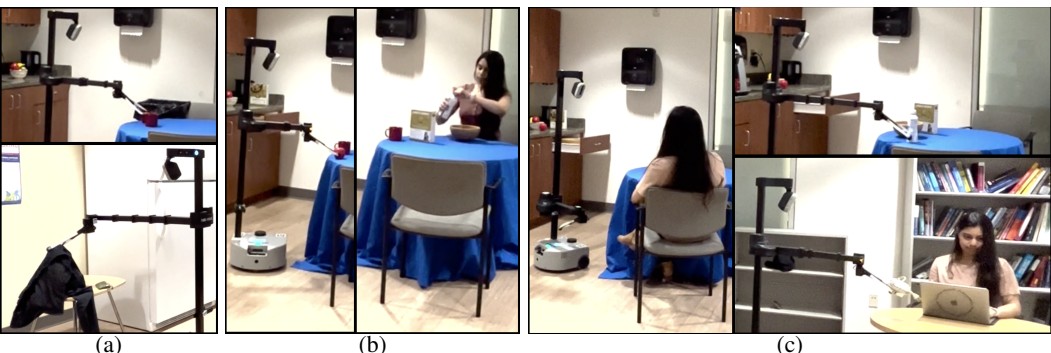

|     (a)     |     (b)     |     (c)     |

Figure 5: (a) A Stretch robot assisting proactively with a user's morning routine, by acting on most likely predictions. (b) These tend to fail when the user chooses a less-frequent variation. (c) By querying the user's about their intent, the robot is able to assist with different variations.

## 8  Limitations and Future Work

Our approach has a number of limitations that present opportunities for future work. First, we do not currently model information about the state of an object (e.g., clean vs dirty plate), and semantic correlations between objects (e.g. spoons and forks are both silverware), which have been shown to be useful in modeling object placement [46, 47, 48, 49]. Second, extending the formulation to a continual learning problem would enable more effective long-term adaptation as user behavior may change over time; the HOMER datasets currently do not model changing user routines. Finally, user-centered factors should be considered and further evaluated to better understand user preferences with regard to types and frequency of assistive robot actions and queries. Our proactive approach can be personalized by changing level of robot assistance through tuning precision v.s. recall, changing amount of active queries through the information gain threshold, and including different types of personalized assistive behaviors in response to different activities.

---

[4]Demo video is available at `https://youtu.be/zLlyM20Bi_8`

[5]This implementation of object mapping does not inventory closed containers and is thus limited to objects directly observable without environment manipulation. However, SLaTe-PRO has no such constraint and can be combined with a different mapping system to overcome this limitation in the pipeline implementation.

**Acknowledgments**

This work is sponsored in part by NSF IIS 2112633 and Amazon Research. In addition, we would like to thank Dr. Weiyu Liu (Stanford University) for technical help regarding model structure and training.

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

# Appendix

## A  HOMER+

We contribute HOMER+ an enhanced version of the HOMER [6] dataset, which is a first-of-its-kind longitudinal behavioral dataset capturing object-interaction level information. We create HOMER+ to emphasize variations in the activity patterns within the simulated households to more accurately model the stochasticity in the real world.

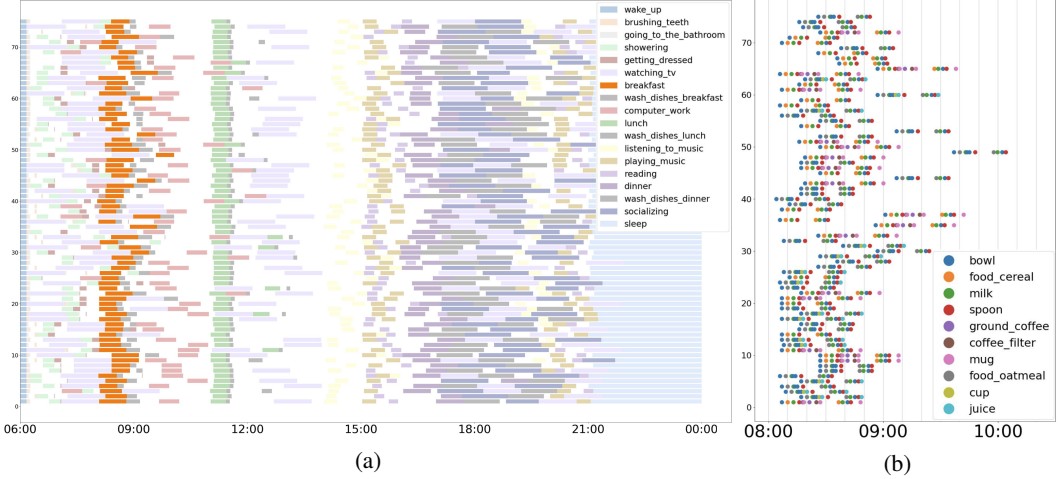

Figure 6: (a) Temporal variations in breakfast activity and (b) corresponding object usage in a simulated household in HOMER+ through breakfast activity over 75 days of data. Notice day-to-day variations in both: start times and objects being used.

### A.1  HOMER: Strengths and Limitations

The HOMER [6] dataset contains weeks-long data illustrating routine user behavior with object-interaction-level detail. It comprises of 22 activities of daily living, and is based on crowdsourced data containing 1) action sequences of how people perform each activity, and 2) temporal distributions representing different habits of when the activity is done during the day. A temporal habit and a script for each activity are used to compose a fictional household, and samples representing a day in the household are generated from the resulting temporal distribution. The resulting dataset presents realistic temporal noise in user activities, since the activities are sampled from a highly stochastic distribution, while maintaining some patterns which the model is expected to learn.

**Weaknesses:** HOMER has no variation within each activity since it picks a single action sequence script per activity. This assumption results in behaviors such as always eating cereal and milk for *breakfast*. It also ignores *sleeping* as an activity, therefore ignoring variations in when the user starts their day in the morning, and ignores correlations between activities of *having meals* and *washing dishes*.

### A.2  Creating HOMER+

To create HOMER+, we manually insert variations in the crowdsourced action sequences. For instance, if the *breakfast* script includes having cereal and coffee, its variations might include eating oatmeal instead of cereal, and/or having juice instead of coffee. This results in different objects being used for breakfasts on different days as shown in Figure 6b. We pick one script per activity, and generate up to 3 variations, which can then be randomly used each time the activity is done. We also insert *going to sleep*, *getting out of bed* and *taking a nap* activities to allow varying start times of the day, and split activity of *washing dishes* into activities associated with each meal to enhance consistency, such that the objects used are the ones washed. Ultimately, this results in a set of 47 scripts encompassing 24 activities. We compile the temporal activity distributions in the same

way as HOMER, and sample routines by randomly picking script variations within each activity. This results in a dataset representing three households, consisting of similar activity scripts across households but enhancing variations within each household, relative to HOMER. This allows us to really test how predictive models perform under such more realistic noisy conditions.

### A.3 Characteristics of HOMER+

The final dataset HOMER+ contains sequences of activity labels and object locations information throughout the day for several weeks, for each of the three simulated households. The activity labels come from the set of 24 activities, with no distinction made based on the variation being performed. This makes the dataset more challenging but maintains a realistic assumption that such nuances are difficult to observe through activity recognition. The activities are sampled from a temporal distribution, resulting in a dataset with natural temporal stochasticity, e.g. in one household, the user has dinner anywhere between 4pm to 7pm, washes dishes right after or hours later, and socializes, plays and listens to music and watches TV in a random order before and after dinner, as shown in Figure 6a. Within each activity the object usage may vary, as shown in Figure 6b for a single activity of *having breakfast*. The temporal variations in activities, cause any given activity to be followed by one of 10 different activities on average. Taking into account the variations we introduce, this amounts to about 20 different activity variations following any activity. This makes the predictive task significantly more challenging, relative to not having such variations. In addition to the multitude of possibilities of semantic activity sequences, predictions on an object level require an understanding of the expansive space of object locations. Each household consists of 93 entities, which all serve as potential locations for the 59 dynamic objects. Thus, there are about $10^{115}$ ($93^{59}$) different potential scene graphs representing object-location combinations, making the full modeling space of probability of a scene graph conditioned on an observed scene graph blow up to $10^{230}$.

## B  Out-of-distribution Scenarios

A robot deployed in home environments will need to deal with novel situations, which might be out-of-distribution for the learned predictive model. In such anomalous situations, we do not expect the robot to provide perfect assistance by anticipating sporadic user needs, but we would want the robot to not take disruptive actions. Semantic patterns in our dataset, and therefore deviations therein, can be interpreted in terms of the activity $a$, repetition of an activity $r$, objects used $o$, locations where objects are moved $l$, and the time when activity occurs $t$. We express deviations in routines through addition $+x$ or removal $-x$ of each of the above variables $x \in \{a, r, o, l, t\}$, and create hand crafted cases representing each deviation. Our model does not include explicit safeguards against such out-of-distribution behavior, but we report the corresponding model responses in Table 1 to qualitatively understand how the model would behave in such circumstances.

Overall, our model does overfit to seen activities, sometimes moving the irregularly used objects back, but it does not make random predictions, such as moving unrelated objects or misplacing objects. Note that our model learns about each object from scratch by representing them through one-hot vectors, so we do not expect generalization to semantically similar objects. However, future work could explore the use of semantically informed representations to overcome this limitation.

| | Anomaly Type | Examples | Model Response |
|---|---|---|---|
| $+a$ | Adding an unknown activity | party, repotting plants, new medication | If novel objects are used in an unseen activity, then the robot ignores their movement, which is the desired behavior. But, if common objects are used in an unexpected way (e.g., using kitchen bowls to repot plants), then the robot tries to return them, which might disturb the user. If the objects are likely to be used around that time at a different location (e.g. bowl at the table at breakfast time) then the robot moves it to where it is expected to be used, otherwise it moves the object where it is usually stored. |

| | | | |
|---|---|---|---|
| $-a$ | Not performing a usual activity | vacation, having an evening out, sick | The robot adheres to routine, continuing to bring things out, and cleaning them up. This may cause some undesired activity and additional reasoning methods should be added to handle such cases. |
| $+r$ | Repeating a usual activity | having lunch twice, watching TV or listening to music multiple times | In cases where certain activity is observed to usually happen exactly once, the robot will not anticipate its second occurrence, but will provide assistance once the user initiates the activity, which is the desired behavior. For instance, if the user decides to have lunch again, it will assist in bringing out other related objects once the user has started the activity (e.g. bring out a plate once the user brings out a pan and oil), and will also help in returning the objects after use. |
| $-r$ | Not repeating an otherwise repeated activity | Not brushing twice, not playing music or working multiple times a day | If the activity is repeated in a consistent manner such as brushing teeth twice at consistent times, the robot would expect them to occur and keep objects ready. If the activity is repeated but not at consistent times, then the robot learns to not try to anticipate their occurrence but limit assistance to returning the objects after use. It maintains this behavior when the activity occurs fewer (or more) times than usual. Note that such repetition anomalies for inconsistent activities are included in the HOMER dataset. |
| $+o$ | Using additional objects in an activity | Having donuts or pizza for breakfast | The robot is not able to associate these new objects with the activity they are a part of. In some cases, the robot returns the novel objects back to where they are stored, and in other cases leaves them untouched. |
| $-o$ | Not using a commonly used object | Not using plates for lunch | If an object is consistently used in the activity (e.g. bowl for breakfast), then the robot continues to bring out that object. If the object is used only in a less frequent variation of an activity (e.g. one of oatmeal or cereal for breakfast), then the robot learns to wait for the user to bring out the object, and if the user decides to not do so, nor does the robot. |
| $+l$ | Using objects at a new location | Having breakfast in bed, working in the dining room | The robot is expecting the same set of objects to be used at a different location and so it assumes they have been misplaced. As a result, it tries to return the objects to where they are usually used, which might cause an inconvenience to the user. |
| $-l$ | Not using objects at the usual location | Not working at the desk, not having breakfast at the table | This is subsumed in the above case since using objects elsewhere includes not using them at their usual location. |
| $+t$ | Performing the activity at a later time | having a late dinner/breakfast | If a consistent activity is performed unusually late, the robot prepares objects at their usual time, but restores them if a long delay occurs. |

| $-t$ | Performing the activity at an earlier time | having an early dinner/breakfast | If a consistent activity happens slightly early such that the usual time of occurrence is still within the robot's proactivity window, then the robot manages to prepare objects in time. However, if the user starts the activity earlier than that, then the robot fails to prepare in advance but does clean up after. |

Table 1: Out-of-distribution scenarios for the HOMER dataset, and corresponding model responses

