# OpenReview forum: "Predicting Routine Object Usage for Proactive Robot Assistance"
_robot-learning.org/CoRL/2023/Conference — CoRL 2023 Poster_

### Official Review · Reviewer_DVTZ · 2023-07-12

**Confidence:** 2
**Originality:** Good
**Technical Quality:** Very Good
**Clarity Of Presentation:** Very Good
**Impact:** 4

**Recommendation:**

Weak Accept: I recommend accepting the paper, but will not argue for my recommendation if the majority of other reviewers have a different opinion.

**Review:**

Although the work seems highly inspired by STOT, it does provide a clear motivation, formulates and presents solutions, and finalizes with experimental evidence on convincing evaluations. The pipeline appears to be highly dependent on other existing methods (Kimera and SparseInst), which makes it a bit questionable how much of the final performance is actually dependent or affected by the performance of those elementary components. The final results are nevertheless, quite impressive. The paper is well-written and easy to follow.



**Quality Of The Limitations Section:**

Limitations are addressed clearly

**Questions For Rebuttal:**

About STOT you said "The resulting model performed well on more consistent user routines, such as using  a plate for dinner, but was unable to predict less consistent activities, such as socializing."
However, I failed to see in this submission if your method predicts social activities. Was there support for that sentence that I simply missed?

Is the location of objects learned in absolute coordinates of the observed scenario? Can the method learn the relative position of objects such that if the subject moves to a new house, the previous relational plan is directly applicable without additional (or perhaps minimal) re-training?

How was the threshold of 0.5 for the information gain tuned? Could you evaluate the user's reaction when the robot decides to make too many questions?

Could you evaluate the user's reaction when the robot ends up doing the wrong task? Is it more frustrating to have a robot that shows too much uncertainty, and makes queries in obvious situations; or a robot that is very assertive but decides to do the wrong actions?



**Robotics Focus:**

Sufficient demonstration on hardware

**Summary Of Paper:**

This paper aims to learn the assistive actions of robots that can proactively engage in tasks based on previous observations of human activities. The paper seems to be highly influenced by the prior work Spatio-Temporal Object Tracking (STOT). However, the paper reveals some of the limitations of STOT and proposes improvements. While STOT was based solely on object information, this manuscript extends that idea with models of user activities and a component of interactive queries. Real-world experiments show that the proposed method outperforms STOT.


**Summary Of Recommendation:**

Despite being incremental w.r.t. STOT, and being dependent on the performance of other methods (Kimera, SparseInst), the motivation and story of the paper is quite clear and convincing. The paper is also well written and evaluations were done to an appropriate level that supports the claims of why the original STOT was enough without human activities and active queries.

---

### Official Review · Reviewer_ymJh · 2023-07-20

**Confidence:** 3
**Originality:** Good
**Technical Quality:** Good
**Clarity Of Presentation:** Very Good
**Impact:** 2

**Recommendation:**

Weak Reject: I recommend rejecting the paper, but will not argue for my recommendation if the majority of other reviewers have a different opinion.

**Review:**


On the positive side, the paper is very well written and mostly clear. The dataset can be a good contribution for people developing assistive technologies.

The method itself is a fairly straightforward combination of existing machine learning techniques, it seems the primary novelty is the ability to seek feedback, though the method for selecting what questions to ask is also based on existing information-theoretic techniques for minimizing uncertainty.

While the work is definitely related to assistive tech and assistive robotics, it is still a bit difficult for me to say that this is a robotics paper rather than an applied machine learning paper. The robot demonstration shows mainly that you can do pick and place which isn't that novel in itself.

The framework does not handle objects that may be unobservable at certain times, e.g., objects in containers, drawers, etc., i.e., the framework assumes that all objects' locations are always known. This is a major limitation to applying this in practice as most objects in a household are covered typically.

At a high level, there needs to be some discussion and ideally a way to address, out of distribution settings at test time, e.g., when something novel is happening (e.g., user is having a party) that was not captured in training data; encoders-decoders generally do pretty bad in novel situations.


Some questions and minor comments:

In paragraph 2 of the introduction, does this mean that you assume the user is still able to perform all activities as for the robot to get observations? what happens when the user is actually disabled and such data is not available or limited to the actions the user can actually perform?

At the end of the introduction when reporting F-measures, at this point, it not entirely clear what the classification task actually is and hence difficult to tell how good or bad these scores are.

First paragraph of section 3, wouldn't the set of locations be continuous? how is an individual location represented?

Second paragraph of section 3, do you assume time is discrete or continuous? If the former, how do you discretize it? Also, is u_t represented as a string or something else?

With regards to encoding time of day, is that absolute or relative to let's say, when the user wakes up, has a dinner, etc? Wouldn't this require the user to do their routine at the same time each day? It is possible a user wakes up earlier or later than usual, or goes home at a different time than usual

When describing the dataset and stating that there are 93 entities+locations, this seems like a very low number of objects and locations for several-weeks worth of data -- were any objects filtered out? There are typically hundreds of objects in a household.








**Quality Of The Limitations Section:**

Additional details required

**Questions For Rebuttal:**

See questions in review.

**Robotics Focus:**

Relevant but unlikely to deploy to hardware in near future

**Summary Of Paper:**

The paper addresses the problem of predicting how users move the objects around them over the course of daily activities and presents an architecture that can also seek user feedback when making such predictions. The method is based on graph-based encoder-decoder network which is trained on each specific user. The method is evaluated on a dataset and also a small demonstration is conducted on a physical robot.



**Summary Of Recommendation:**

My recommendation is due to the following:

1. There are major limitations to the representations and assumption that would make this difficult to deploy in practice.
2. The paper is mostly an applied machine learning paper rather than a robotics paper.

---

### Official Review · Reviewer_4vMT · 2023-07-20

**Confidence:** 4
**Originality:** Very Good
**Technical Quality:** Very Good
**Clarity Of Presentation:** Very Good
**Impact:** 3

**Recommendation:**

Strong Accept: I recommend accepting the paper and will argue for my recommendation even if other reviewers hold a different opinion.

**Review:**

I am enthusiastic about the approach the authors present as it is close to a maximal use of the information that is readily available to an observant robot (where objects are and what people are doing, over time). At a high level, I believe this paper gives us an indication of how well this kind of proactive assistance can work, at least quantitatively; we should expect it to act appropriately only about half the time. The exact picture of what we should expect is obscured by using F1, because we can imagine situations where either precision or recall are more important. In an assistive use case, we may prioritize recall to reduce the burden on the user. In other settings, users may be perturbed by the robot moving an incorrect object (if for instance, they notice that something has moved without apparent cause, and are worried). While breaking out precision and recall would help, what this really indicates is that the questions readers are likely to have are now largely about the qualitative picture for this kind of assistance, something which is squarely future work.

**Quality Of The Limitations Section:**

Limitations are addressed clearly

**Questions For Rebuttal:**

Figure 2. What quantity do the error bars represent?

Figure 2b. Why does SLaTE-PRO show lower variance and improved accuracy as the horizon extends?

**Robotics Focus:**

Sufficient demonstration on hardware

**Summary Of Paper:**

This paper proposes a new model for the task of predicting routine object usage, which has a direct application in enabling a robot to rearrange objects proactively to assist users. The model extends prior approaches (which predicted solely on the basis of previous object locations) by including historical object locations and activity recognition information. Intuitively, variations in a user’s activities are likely to lead to variations in the objects they interact with. Further, the authors incorporate a query mechanism which enables the robot to ask for a next activity or next object when the model has no confident predictions. The new method is compared against previous work on an enhanced dataset, and results show that the enhancements lead to better prediction accuracy.

**Summary Of Recommendation:**

This work addresses important shortcomings of existing approaches, and does so with good clarity and reasonable experiments. This paper will inform and motivate future work in designing assistance systems.

---

### Decision · Program_Chairs · 2023-08-30

**Decision:**

Accept (Poster)

**Comment:**

This paper proposed a system that  learn to anticipate user's needs from past observations and use the prediction to provide proactive assistance.
While the work is highly inspired by STOT, it applies the method in a new and unique setting. The paper provides a clear formulation and method. The experimental evidence is convincing and informative. Overall, it is a valuable contribution to the field.  The reviewer points out a few limitations of the proposed approach (unobservable objects, out-of-distribution settings). AC encourages the author to include these discussions in the paper and clearly describe what are the remaining issues that could be addressed by future work.